# A Study on Additive Manufacturing for Electromobility

Dirk Schuhmann [1],* , Christopher Rockinger [1] , Markus Merkel [1] and David K. Harrison [2]

1   Institute for Virtual Product Development, Aalen University, Beethovenstraße 1, 73430 Aalen, Germany
2   School of Computing, Engineering and Built Environment, Glasgow Caledonian University,
    Cowcaddens Road Glasgow, Glasgow G4 0BA, UK
*   Correspondence: dirk.schuhmann@hs-aalen.de

**Abstract:** Additive manufacturing (AM) offers the possibility to produce components in a resource-efficient and environmentally friendly way. AM can also be used to optimise the design of components in mechanical and physical terms. In this way, functionally integrated, lightweight, highly efficient, and innovative components can be manufactured with the help of additive manufacturing in terms of Industry 4.0. Furthermore, requirements in the automotive industry for drivetrain components are increasingly being trimmed in the direction of efficiency and environmental protection. Especially in electromobility, the topic of green efficiency is an essential component. Exhaust emission legislation and driving profiles for evaluating vehicles are becoming increasingly detailed. This offers the potential to apply the advantages of AM to vehicle types such as conventional, utility vehicles, and nonroad mobile machinery (NRMM), independent of the electrical drivetrain technology (hybrid or fully electrical). AM also allows for us to produce optimally adapted components to the respective requirements and use cases. In this review, the intersections of AM and electromobility are illuminated, showing which solutions and visions are already available for the different vehicle types on the market and which solutions are being scientifically researched. Furthermore, the potential and existing deficit of AM in the field of electromobility are shown. Lastly, new and innovative solutions are presented and classified according to their advantages and disadvantages.

**Keywords:** electromobility; additive manufacturing (AM); metal 3D printing; selective laser melting (SLM); lightweight; modular and scalable; function integration

## 1. Introduction

Due to the ever-increasing demands for climate protection, and the requirement to be less dependent on fossil fuels, the development of electric drives is advancing at a rapid pace. Electromobility is globally regarded as the key to climate-friendly mobility. Therefore, AM is used to efficiently support this technology and allows for developing components with a high degree of design freedom. The properties of the highly efficient parts produced by AM are characterised by load-optimised bionic shapes with very thin walls and lattice structures. Furthermore, there is the possibility of using a wide variety of materials optimised for the process and functional integration. This helps in efficiently complementing electromobility. Many automotive manufacturers have understood the potential of AM, and have successfully integrated it into the development and manufacturing process (e.g., spare parts, prototypes, component production), which is evidenced by a rising number of press announcements and amount of research activity [1–84].

In addition, AM machine manufacturers are trying to further develop their machines in the direction of series production [16,17]. There are still various problems to be discussed and resolved. One example is the extrusion force during the manufacturing process [2]. As a result, the selected components of industrial users can be manufactured in series production [18–20]. Powertrain topologies are also constantly being improved and elaborated [1]. Both drive components, such as electrical machines or power electronics, and mechanical axle parts play a role. Especially in the area of axle suspension and suspension,

in combination with wheel-near drive vehicles, there is still potential for future efficient drive platforms [3]. For decades, manufacturing and design have been based on established manufacturing processes that strongly influence and limit the design possibilities and component geometry depending on the process.

Therefore, this study aims to show how far AM positively influences electromobility and the gradual change in the technological development of electric drive topologies such as hybrid and battery-electric vehicles with their potential and deficits. For this purpose, the current state of development and research, and the state of the art are explored and summarised. With this summary, the potential for future electrified drive platforms is shown with the combination of additive manufacturing.

## 2. Materials and Methods

The authors conducted a structured and extensive literature search to ensure the high-quality literary research and identification of existing works, and the possibility of building on and further deepening existing research. Several databases were used for this purpose. On the one hand, ScienceDirect and EBSCOhost and, on the other hand, Google Scholar as the superordinate search engine. The systematic literature search with the keywords used is shown in Figure 1.

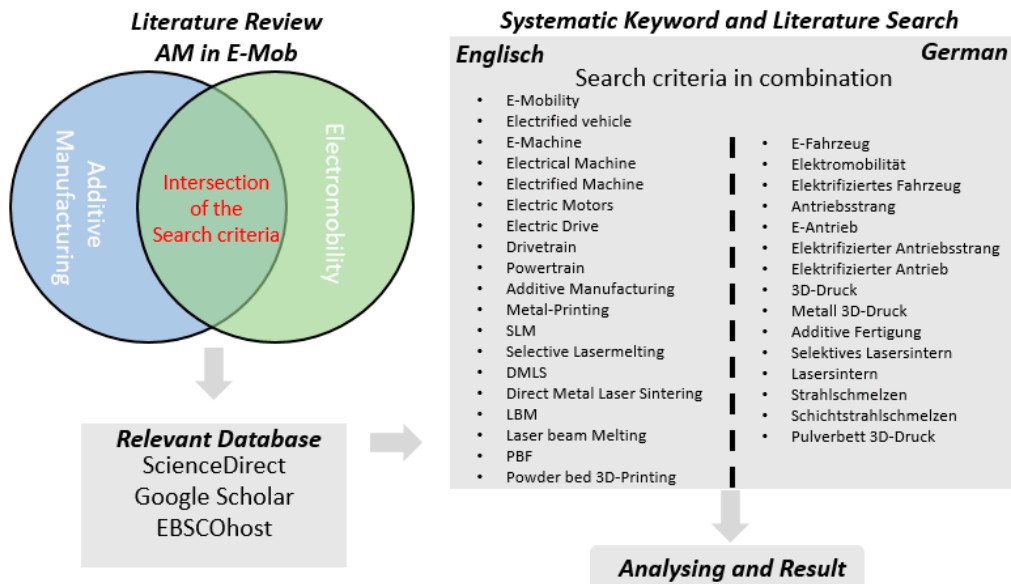

**Figure 1.** Methodical literature search.

For the literature search, we attempted to find an intersection of the generic terms additive manufacturing and electromobility. Additive manufacturing includes several possibilities of manufacturing processes and materials [21]. In view of the number of manufacturing processes and the areas of application of additive manufacturing, the authors were limited to 3D metal printing. Electromobility is a wide-ranging generic term that includes a wide variety of drivetrain topologies. Electromobility includes all types of vehicles with additional electric drives (hybrid vehicles) or fully electric drives (pure e-vehicles); fuel cell vehicles are also included in electromobility [22–25].

Due to the large number of studies, attention was paid not only to the selected search parameters in the form of terms, but also to the novelty of the literature. More than 350 sources were found and evaluated. For this purpose, the abstracts were read, and the contents were compared. From these preselected literature search hits, >150 sources were evaluated and read in greater depth. The articles judged to be of scientific and literary value for the paper were further evaluated according to previously defined criteria and further narrowed down. The literature review shows that there tended to be more grey literature than scientific contributions. Thus, approx. 60% of the search hits were grey literature, and

approx. 40% of the search hits were scientific. This evaluation is based on the guidelines established by the authors. For this purpose, a differentiation was performed as to whether a source was a scientific journal or an Internet source, journal, article, or homepage. The literature evaluation is shown graphically in Figure 2.

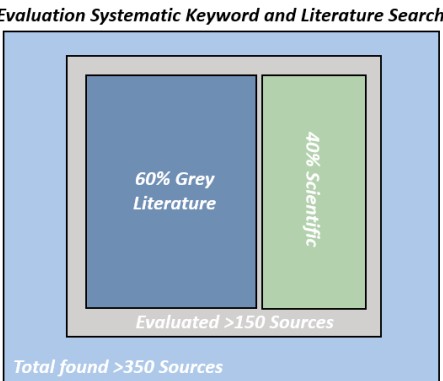

**Figure 2.** Evaluation of the literature search.

The diagram shows the total number of literature hits with >350 sources. Of these, >150 sources were useful. The evaluated literature consisted of approx. 60% grey literature and 40% scientific literature.

AM, also known as 3D printing, is the layer-by-layer construction of a component. From a digital 3D model, slicing software is used to divide the component into individual layers that reflect the layer data of the geometry with a layer height of 20–100 μm. Among the seven different AM processes shown in Figure 3, powder bed fusion technology with a subordinate SLM process is the most widely used process in the manufacture of components for electromobility [26–29].

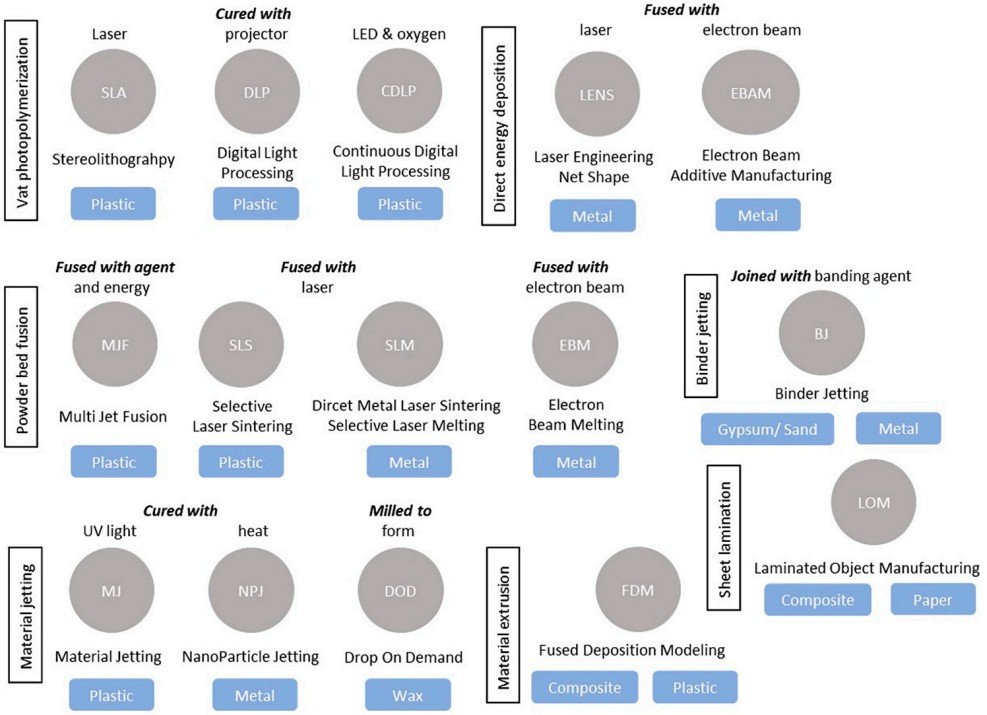

**Figure 3.** AM technologies.

Figure 4 shows the schematic setup of the manufacturing process of an SLM machine. For layer-by-layer manufacturing, an interplay of powder feeding, local powder feed, and the local melting and lowering of the base plate is mandatory. Thus, the component is built up layer by layer through the local melting of the metal powder. In the first step, the powder is applied to the base plate, and the initial layer of the component is melted by the laser. Next, the base plate is lowered, and another layer of powder is applied and melted; this is repeated until the part is converted according to a dataset. In addition, AM parts usually have a density of <99% [30,31].

**Figure 4.** Schematic setup of an SLM machine [32].

Electromobility means that a vehicle is fully or partially electrically powered. The spectrum ranges from fully electric vehicles to those in which the electric drive contributes to the drive to varying degrees; this type of drive topology corresponds to hybrid drives. Depending on the degree of hybridisation, we speak of micro, mild, full, and plug-in hybrids. Figure 5 gives an overview of a plug-in hybrid drivetrain [33–35]. The drivetrain of the vehicles is almost identical for a hybrid vehicle as well as for a fully electric vehicle. The components for the electrification part are usually only different in terms of wiring and dimensions. The main components are the traction battery, electric motors, and power electronics. The mechanical drive train is comparable to that of conventional vehicles. However, the transfer case and the drive shaft are usually missing in four-wheel-drive vehicles. Instead of these components, separate motors are often installed for the front and rear axles [22,23,36,37].

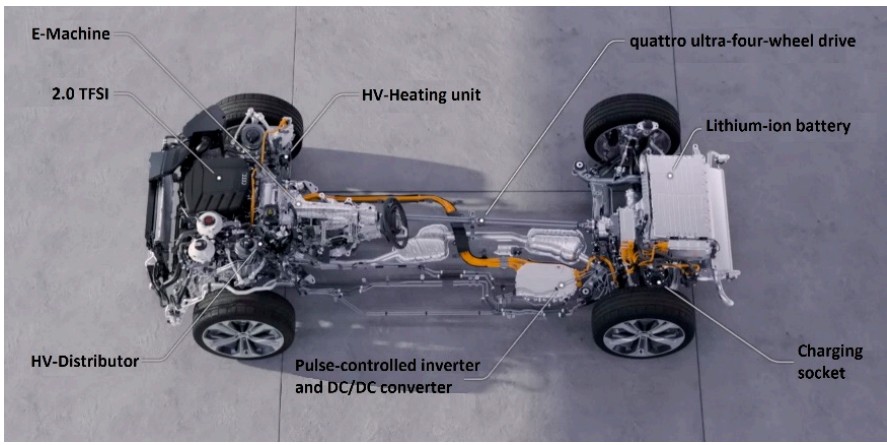

**Figure 5.** Audi Q5 55 TFSI e quattro—PHEV drivetrain [38].

We categorised the drivetrain with regard to additive manufacturing. This categorisation allows for more clearly presenting the evaluation of the found literature. This categorisation is shown in Figure 6.

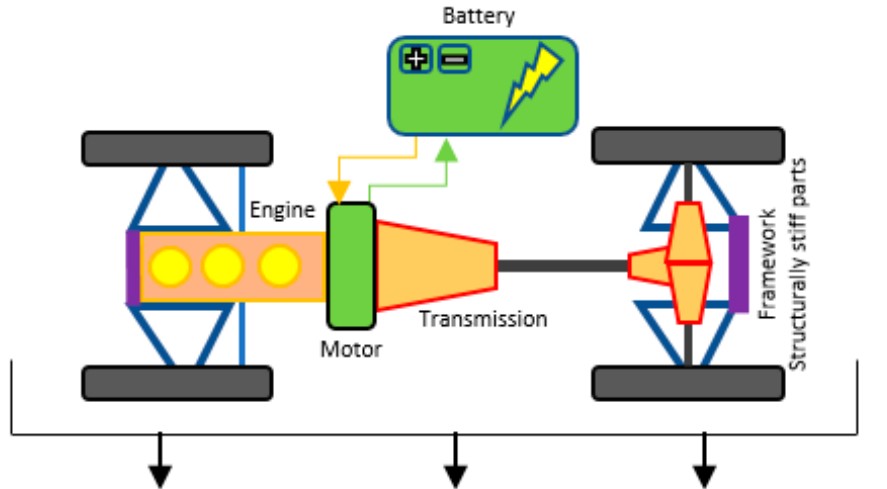

**Figure 6.** Drivetrain categorisation with regard to additive manufacturing.

For the categorisation, the drivetrain was evaluated and assessed with regard to its individual components. In summary, the drivetrain components can be divided into the categories of mechanically stressed components, functional components, and "thermally stressed components". Overlapping categories are possible, such as mechanically stressed functional components. These overlaps are not listed separately.

## 3. Trends of AM for Electromobility

The development of drivetrains for electromobility offers far more potential than simply replacing the combustion engine with an electric motor. With the possibility of power-specific AM design, and the resulting wide variety of electric machine designs, new drivetrain topologies can be designed. The gradual reduction in mechanical drive components improves the overall efficiency of the drivetrain (tank to wheel). With regard to metal 3D printing, it is possible to combine components, functionalise them, and design them using lightweight construction approaches, so as to incur weight and component savings. The literature research showed that the degree of maturity of AM technologies in the field of electrical machines is still relatively low, and the used components often have poorer physical properties than those of their equivalents from proven manufacturing processes. In contrast, however, the structural components show a high degree of maturity. In addition, the practical industrial implementation of AM seems to be far ahead of research in some areas. In most cases, research is mainly limited to single components and materials. In a few cases, a complete approach is targeted [12,39].

### 3.1. Materials

Additive manufacturing in the form of 3D metal printing offers the possibility to use a wide range of metal alloys and allows for the production of the desired materials to achieve the wanted part properties. The different materials are displayed in Figure 7a. Figure 7b shows the mechanical properties of the various materials' processing. These values are comparable, for example, to the properties of castings. The nomenclature in Figure 7b refers to either the chemical composition of alloys such as AlSi10Mg or the material number according to DIN [40–43].

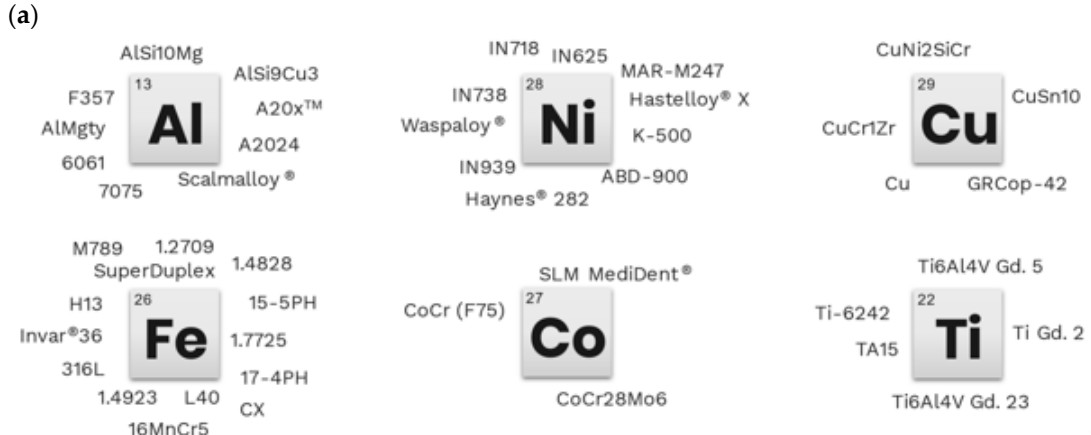

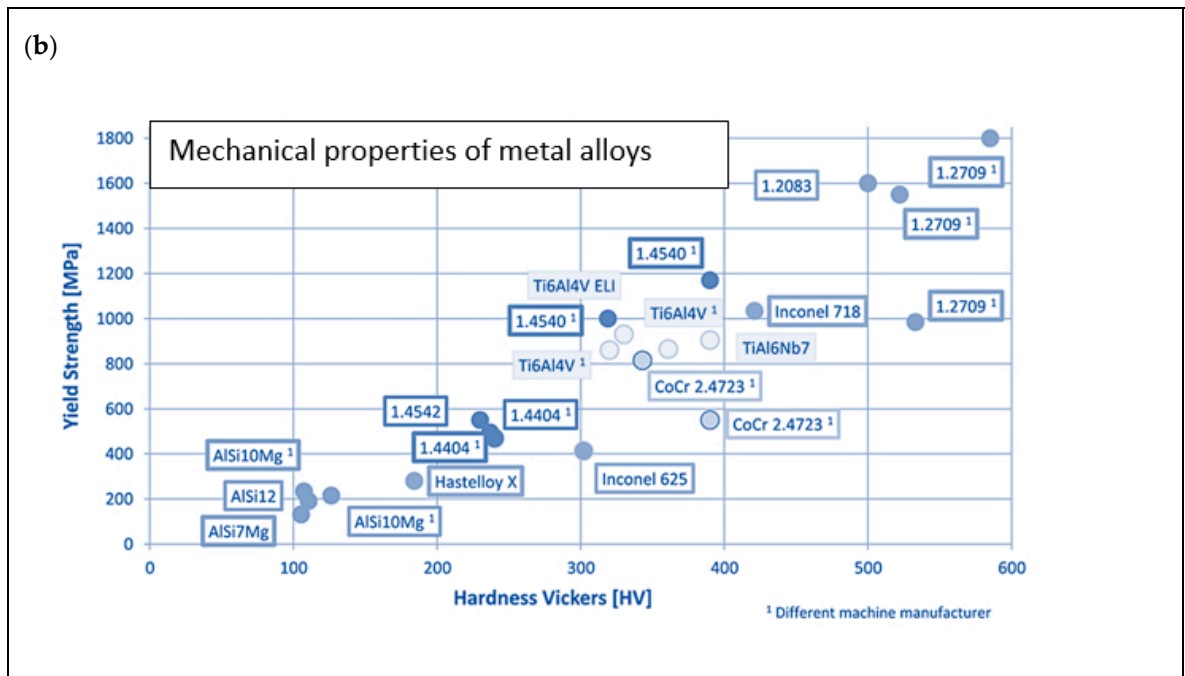

**Figure 7.** (**a**) Available metal powders [40]; (**b**) mechanical properties of metal alloys (AM powder) [27].

### 3.2. AM According to a Drivetrain Hybrid/Battery Electric Vehicle

Due to the steadily increasing vehicle weight resulting from electrification and the associated deterioration in driving dynamics, it is of great interest to integrate lightweight construction approaches and innovative highly efficient components into the vehicle [44–46].

A drivetrain for battery-electric and hybrid drive topologies can be divided into three component categories with regard to additive manufacturing, as already described in detail in Section 2:

- mechanically stressed components;
- functional components;
- thermally stressed components.

Intersections in the categories and combined total solutions are also possible.

### 3.3. Functional and Thermally Stressed Components for Electrical Machines

The main subassemblies of electrical machines manufactured with AM, also considered in Section 1, are: motor housings, elements for thermal management, coils/windings, and stators/rotors.

Examples of machine housing for applications involving electric vehicles (EVs) are shown in Figure 8. Hardware- and software-based CAD models are displayed there. In these four cases, a special integrated cooling channel or geometry is designed for liquid cooling directly into the housing to improve efficiency and enhance the overall functionality in terms of thermal management.

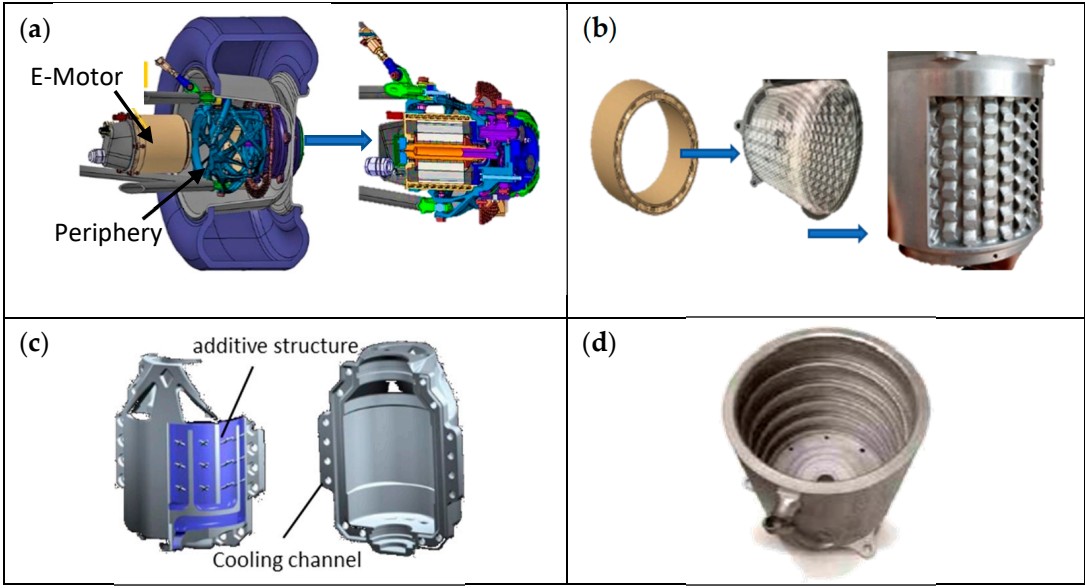

**Figure 8.** (**a**) Electrical motor mounted in wheel hub; (**b**) electrical motor housing with integrated cooling [47,48]; (**c**) housing with integrated cooling [49]; (**d**) housing with helix structured cooling channels [50].

The housing shown in Figure 8a was redesigned to be manufactured with AM in AlSi10Mg because of the critical temperature conditions under which the original motor had to operate. As shown in Figure 8b, AM allowed for a complex design for integrated cooling. The redesigned cooling geometry resulted in better heat dissipation, and weight and operating-pressure reduction. This increased the overall heat conduction and efficiency of the cooling system by nearly 20%. In addition, increased temperatures at peak loads during operation are quickly cooled down [47,48]. Figure 8c also shows another CAD option of integrated cooling [49]. Figure 8d shows an electric motor with a helix cooling channel inside, realised as a single component. The application in practice shows that the cooling capacity can be increased by up to 37%, while the weight is reduced by 16% [50].

Another example is different kinds of housings or cooling systems, presented in Figure 9. AM was successfully used to manufacture housing from a single component with a functionally integrated cooling channel and additional connection points; see Figure 9a. The housing protects the power electronics during operation and assembly. Furthermore, approaches regarding modularity can more easily be implemented [51].

In [52], the authors investigated AM heat guides (HGs) produced from AlSi10Mg that are mounted between windings in an actively cooled stator. The study showed that HGs could be fabricated using AM, and their complex geometry had reduced mass, low power dissipation, and good thermal conductivity. The initial results using FEM temperature analysis showed better handling of the input power by 40% in low-frequency operation, and an improvement of 20% in high-frequency operation. Theoretically obtained data were experimentally validated and showed good correlation with the theoretical measurements. During the measurements, the components had comparatively high electrical resistance, but this could be neglected in terms of heat dissipation. In general, this improved the thermal conductivity between winding and stator by 55–85%.

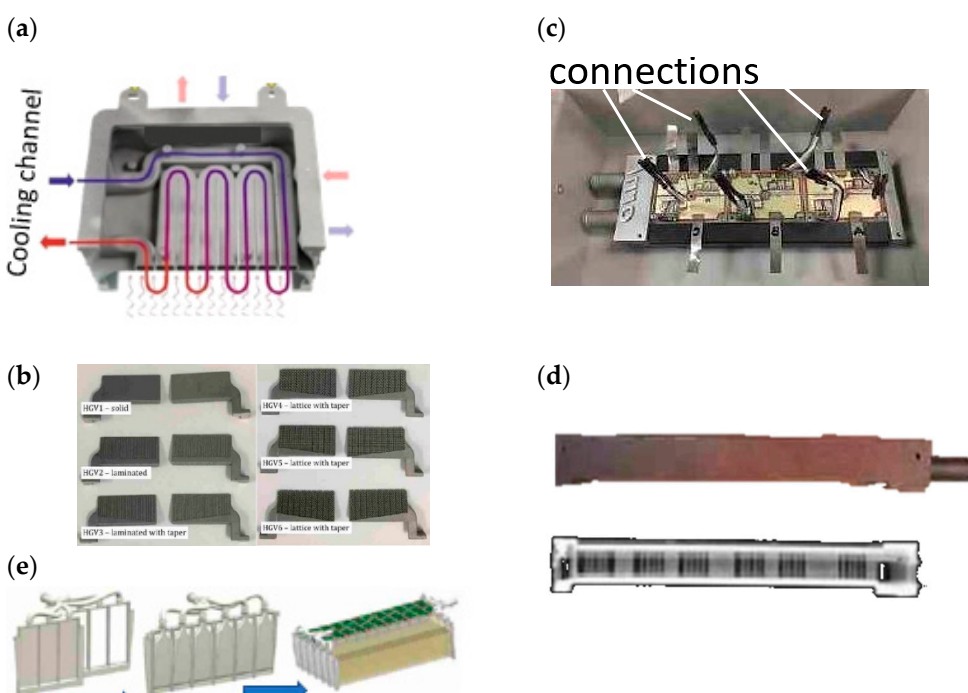

**Figure 9.** (**a**) Multifunctional housing for power electronics [51] (**b**); heat guides for an electrical machine [52]; (**c**) inverter with cooling plate; (**d**) X-ray view cooling plate [53]; (**e**) battery cooling system [54].

An inverter's cooling system, shown in Figure 9c, was fabricated from the aluminium alloy AlSi10Mg using AM and then examined. Figure 9d shows the inverter and heat sink with an X-ray image to illustrate the internal geometry. The result was the 99% efficiency of the inverter under several operating conditions. Moreover, an increase in power density was possible [53].

The optimised additively manufactured battery cooling components shown in Figure 9e enable the maximal usable power to be used up at any time during operation. In addition, the battery can be preheated to 45 °C before start-up. The cooling components had a total weight of 225 g [54].

A lightweight rotor design approach is also helpful in further reducing and optimising the mass of an electric machine, as illustrated in Figure 10 with two examples. By using AM, the rotor displayed in Figure 10a could be built according to a lightweight design concept, as can be seen from the sectional view of the CAD model. Furthermore, the efficiency of the rotor could be enhanced and validated through the experimental investigations on the physical model shown in Figure 10b [55–58]. Figure 10c successfully demonstrates how a rotor could be realised by using lattice structures due to an efficient lightweight construction approach. Tool steel (H13) was used for the rotor itself. Although the magnetic properties of the tool steel (H13) are relatively poor, the properties of existing soft magnetic composites (SMCs) were achieved with heat treatment. Compared to a conventionally manufactured component, the total mass of the rotor was reduced by 25%, thereby reducing the moment of inertia by 23% [59].

Following the presentation of housing and attachments, windings are one of the most important components of an electric motor, which are usually produced from copper due to its relatively high electrical conductivity. The fact that some parameters in the winding production and design complicate the manufacturing process, AM allows for new design approaches, and improves the performance and manufacturability compared to conventional methods of design and manufacture [61]. As shown in Figure 11, additive manufacturing allows for improving the power density of motors through the special arrangements of the windings and hairpin manufacturing. The motors can be smaller and

lighter due to the windings, which saves space and weight [61,62]. The better internal cooling of the motors directly in the winding can also be realised. For the windings shown in Figure 11a, the priority was to exploit the geometric design latitude associated with AM and thus reduce ac loss. In this way, it was possible to develop an optimised geometry that minimises losses and significantly reduces the component size. In addition, further studies were published in which the systematic redesign in the form of a highly concentrated winding topology was demonstrated. This resulted in a significant loss reduction of 40% at a hot spot temperature of 145 °C compared to the pervious iteration.

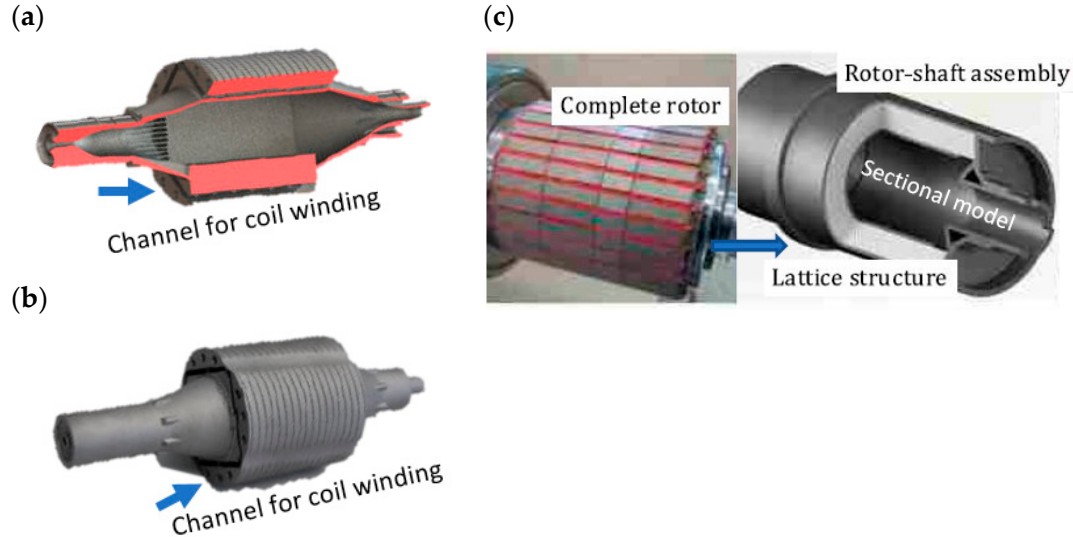

**Figure 10.** (**a**) Sectional rotor view; (**b**) 3D-printed functional model [55–58]; (**c**) rotor with lattice structure [59,60].

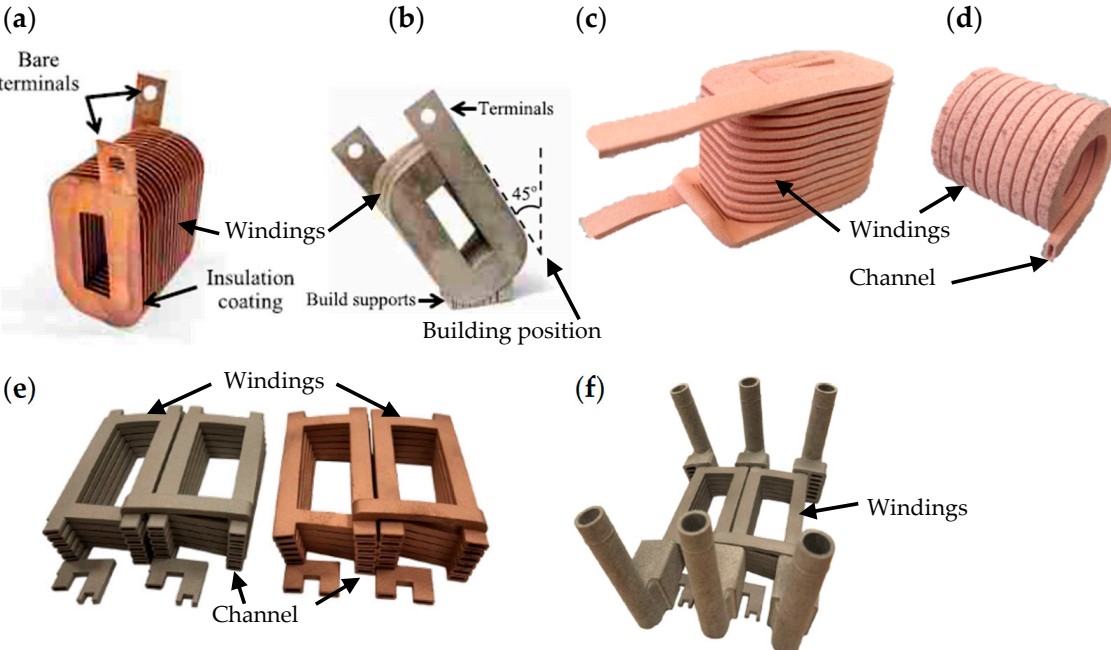

**Figure 11.** (**a**) AM-shaped profile windings; (**b**) self-supporting AlSiMg alloy sample [63–65]; (**c**) 3D-printed copper coil; (**d**) hollow core design [66] (**e**); AlSi10Mg and CuCr1Zr coil samples (**f**); coils with heat exchangers [67–71].

Additive manufacturing allows for improving the power density of the motors through special arrangements of the windings and hairpin manufacturing. The motors can be smaller and lighter due to the windings, which saves space and weight [61,62]. The better internal cooling of the motors directly in the winding can also be realised. For the windings shown in Figure 11a,b, the priority was to exploit the geometric design latitude associated with AM, and thus reduce AC loss. In this way, it was possible to develop an optimised geometry that minimises losses and significantly reduces the component size. In addition, further studies were published in which systematic redesign in the form of a highly concentrated winding topology was demonstrated. This resulted in a significant loss reduction of 40% at a hot-spot temperature of 145 °C compared to the previous iteration [63–65]. Figure 11c,b show that the hollow demonstration windings had the highest conductivity with a resistance of 3.19 µΩ-cm (54% IACS). Due to the discrepancies between the CAD values and the real available parts, it was difficult to determine the exact cross-sectional geometry for resistivity calculation. Furthermore, it could be verified that the material used (pure copper) has enough potential to be used in further applications [66].

The coils from Figure 11e were compared on the basis of their materials (AlSi10Mg and CuCr1Zr). Generally, the choice of material favours aluminium alloys due to their high efficiency and specific power at lower conductivity. The use of CuCr1Zr is still limited because the manufacturing parameters regarding wall thickness and the distance between the layers are not sufficiently known. These limitations hinder the loss reduction of hollow conductors under high excitation frequency. The AlSi10Mg components have the same quality at room temperature as that of the cast aluminium. However, the parts differ depending on the used heat treatment. The electrical conductivity of preannealed components is orientation-dependent. Components that have received T6 heat treatment are less orientation-dependent. The highest electrical conductivity is achieved in build orientation. In this way, the heat treatment, material choice, and orientation allow for the flexible adaptation of the design to the application, improving the machine's efficiency and performance. Figure 11f shows the investigated coils assembled with the heat exchangers [67–71].

*3.4. Mechanically Stressed Components*

As with the lightweight design approaches for electrical machines, the automotive industry is increasingly using additively manufactured lightweight aluminium structural components, as shown in Figure 12, which are intended to help in further reducing vehicle weight. In most cases, this is achieved by optimising the topology to achieve the required stiffness properties while reducing weight.

The topology-optimised seatbelt-buckle bracket was developed using metal 3D printing. In this way, eight components could be realised into one. The component was 40% lighter and 20% more resilient than that of the previous assembly (Figure 12a) [10].

Figure 12b shows a steering knuckle separately produced from AlSi10Mg and assembled. The challenge was to develop the lightest possible component with the greatest possible stiffness in order to keep the origin mass of the vehicle as low as possible. Too high a mass would result in poor spring-damping behaviour. The AM and the design enabled the mass of the component to be reduced by 35% (660 g), and the stiffness to be increased by 20% [72].

The topology-optimised aggregate carrier illustrated in Figure 12c was subject to a wide range of requirements, such as supporting the drive torques and transmitting the vehicle acceleration to the electrical machines. Furthermore, the components are an important element of the acoustic transmission path from the electrical machine and the transmission. In addition, they serve as a support in the event of a rear-end crash. Therefore, AM was used to produce a topology-optimised component that could meet the various requirements. The component was tested in extensive test series. In total, an acoustic improvement and a weight advantage of 1 kg per component (25% weight saving) could be achieved. A sand casting/hollow construction was referenced as a comparison [73].

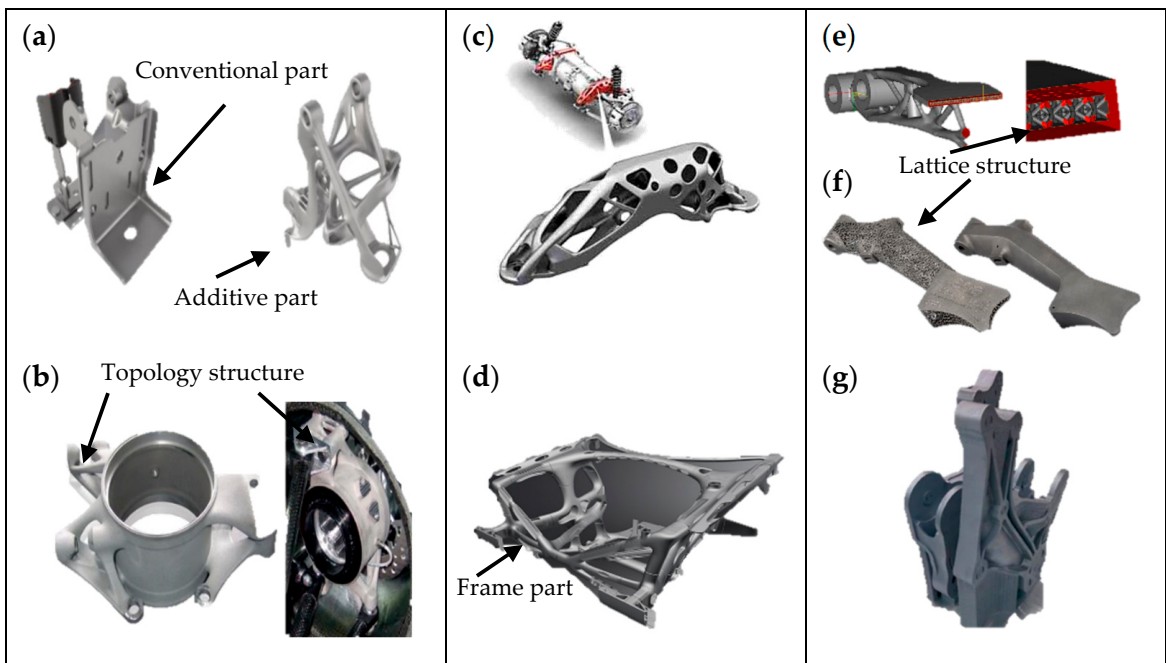

**Figure 12.** (**a**) Topology-optimised multifunctional anchor bracket [10]; (**b**) aluminium steering knuckle [72]; (**c**) topology-optimised aggregate carrier [73]; (**d**) functionally integrated vehicle front structure [74]; (**e**) topology-optimised brake pedal with lattice structure [75]; (**f**) lightweight brake pedal [78]; (**g**) additively manufactured topology-optimised adapter for EVs [76,77].

The front structure concept displayed in Figure 12d shows the additively manufactured part from Scalmalloy. The design was applied according to real load cases in the automotive industry. In addition, it was designed to meet defined crash and vehicle safety requirements. The result was a lightweight front structure with integrated cooling channels and cooling fins, and the potential to integrate actuators and sensors directly into the component [74].

The brake pedal was created utilising Iso-XFEM and lattice structures, and was manufactured with Ti–6Al–4V that had been selectively laser-melted. In order to lessen the severity of the high residual stresses that occur during the printing process, the thickness of the pad was increased, and the hollow area was filled with a body-centred cubic lattice structure because of its self-supporting cell structure. Overall, it lowered the volume, and increased the stability of the design shown in Figure 12e [75].

The lightweight Ti64 brake pedal produced by AM, as depicted in Figure 12f, weighed 22 g (11.6%) less while being more rigid compared to the weight of the previous model (190 g).

The development of a scalable and modular component as an adapter element for changeable integration into the overall axle drive system was the main goal in Figure 12g. The SLM manufacturing process was then used to create the adapter element. A study that illustrates the design process with regard to integration on the vehicle side and implementation with regard to production was the end result [76,77].

The Czinger 21C hypercar illustrates how AM can help in improving vehicle efficiency and performance. The vehicle comprises over 350 AM components, some of which are shown in Figure 13a. All of these components were implemented throughout the vehicle from the frame to the control (Figure 13b), braking (Figure 13c), suspension (Figure 13e), and exhaust systems. This resulted in component mass reductions averaging 15–20% or more in individual cases. In addition, the total mass of the vehicle consisted of 20% AM components. The two major applications that function as the main front and rear impact structures are presented in Figure 13e,f [79,80]

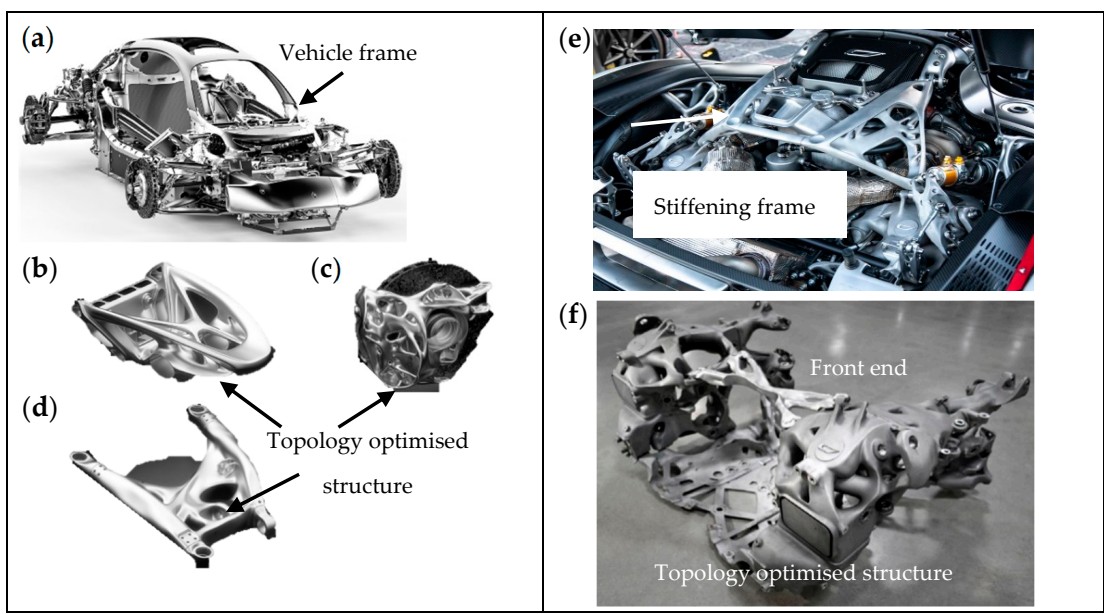

**Figure 13.** (**a**) Czinger 12C; (**b**) steering column; (**c**) brake-node brake component; (**d**) rear lower control arm; (**e**) engine bay; (**f**) Rear structure assembly [79,80].

3.4.1. Mechanical Function Integrated Parts and Total Approaches

As shown in Figure 14, function-integrated and complete solutions produce components with as few individual components as possible, thus reducing assembly effort and implementing components more efficiently with the design freedom of AM. The result is a space-saving application with the same functionality.

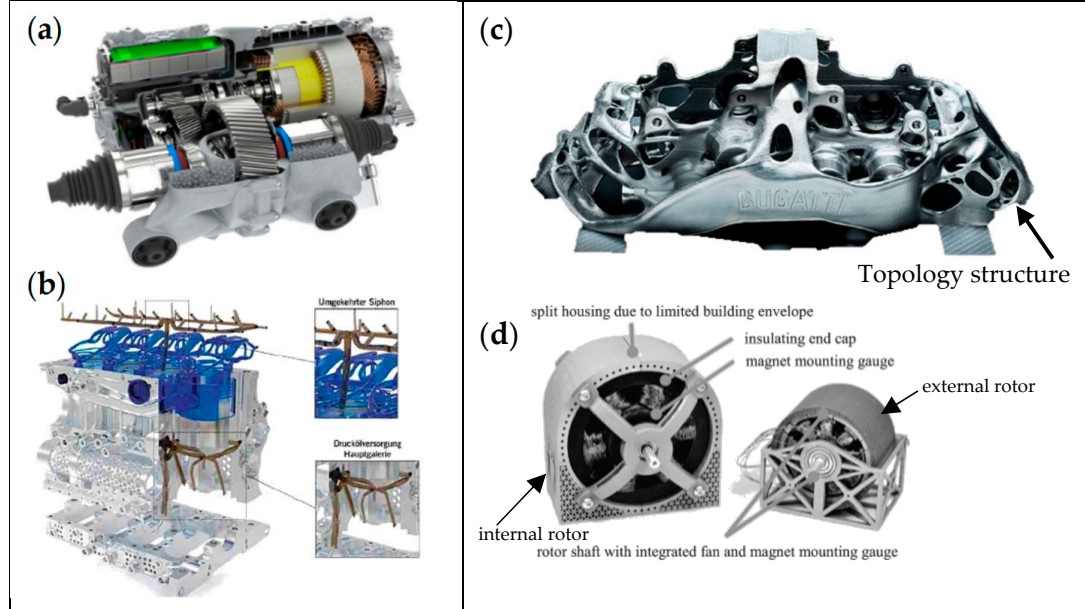

**Figure 14.** (**a**) AM-printed drivetrain unit [39]; (**b**) lubrication and cooling system of the internal combustion engine [12]; (**c**) Bugatti brake calliper [8]; (**d**) 3D-printed electrical drives [81].

Figure 14a shows the 3D-printed prototype of a Porsche e-drive housing. With the prototype, several development stages could be implemented in one manufacturing process, and the assembly effort was reduced by around 40 work steps, and the weight of the housing components was reduced by 40% through the adaptation with regard to function

integration and topology optimisation. So, it was also possible to integrate the transmission heat exchanger, which additionally improved the cooling of the entire drive. The entire e-drive experienced a weight reduction of around 10% due to the lattice structures. Due to the use of lattice structures, the housing had a continuous wall thickness of 1.5 mm, which increased the stiffness by 100%. A honeycomb structure designed on the outside of the drive ensured less vibration of the thin housing walls and significantly improved the acoustics [39].

In the LeiMot Project, the weight of the cylinder head and crankcase was reduced by around 21% compared with a VW diesel engine. Furthermore, in addition to weight reduction, there was potential for efficiency gains. The latter could be achieved by reducing coolant and oil-pump capacity, reducing friction in the piston/liner assembly, reducing emissions during cold starts, and increasing the turbocharger turbine output by insulating the exhaust ports. A variety of AM approaches are used in the LeiMot concept. For example, internal cooling and oil ducts, lattice structures for mechanical use, and as flow elements, geometrically complex stiffening ribs, and integrated insulation systems (Figure 14b) [12].

The function-integrated design of the Bugatti brake calliper (Figure 14c) was based on bionics and produced with titanium (Ti6Al4V). This enabled the topology-optimised component to save 40% (2 kg) of mass compared with its aluminium predecessor, while at the same time offering greater load-bearing capacity [8]. Figure 14d illustrates the 3D-printed and optimised electrical drive. The number of used parts, and the weight of the housing and the bearing shield, were decreased in the feasibility analysis of the 3D printed electrical machine. Additionally, the integration of geometric features that would have not been conceivable utilising traditional manufacturing processes was possible. For better cooling, a fan wheel was incorporated inside the stator. Additionally, there was functional integration in the stator in the form of cooling channels [81].

### 3.4.2. Mechanical Function Integrated Gearboxes and Gearings

Using the SLM method, the topology-optimised gear housing shown in Figure 15a was created using AlSi10Mg. The majority of the article discusses topology optimisation, which was followed by a number of iterative processes before the final design was created. This was especially modified for additive manufacturing. In comparison to its cast equivalent, the resultant package geometry is 40% lighter, 10% more flexible by 98 percent safety. It combines lightweight design principles with improved oil management and load transfer efficiency. Additionally, the chosen 3D-printing method was unstable for components of this scale. Tests demonstrated that the gearbox satisfied the thermal specifications for standard gearboxes [82].

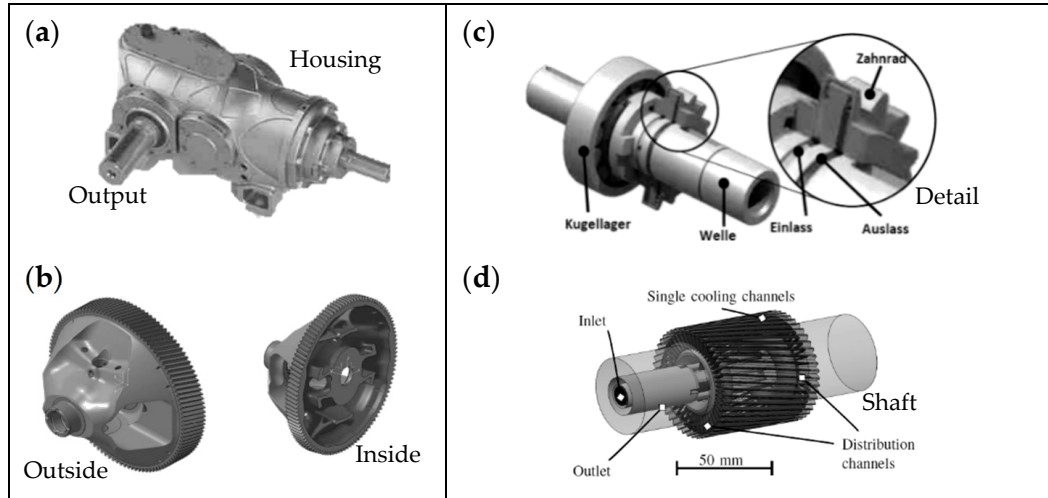

**Figure 15.** (**a**) Gearbox housing [82]; (**b**) topology optimised differential with internal structure [83]; (**c**) gear/shaft assembly with cooling ducts [84]; (**d**) pinion shaft with integrated cooling contours [85].

For the topology-optimised differential of Figure 15b, the AM process parameters for the case-hardened steel 20MnCr5 were determined by detailed static test design, microstructural analysis, and mechanical tests. In addition, great importance was attached to keeping the residual stresses of the additively manufactured components as low as possible in order to minimise the effort required for CNC postprocessing. The case hardening process was developed by considering the standard requirement for surface hardness and penetration depth. Through the use of tensile testing and hardness tests, it was feasible to obtain knowledge of mechanical properties, in this case, with respect to component pressure alignment, the heating of the metal powder during manufacture, stress-relief annealing after production, case hardening, and surface quality. A reference gear was used in order to provide comparison values. Using this high-performance material and the collected data, it was possible to manufacture the previously impossible design solution of a differential including a ring gear. This combination lead to a weight reduction of 13% (1 kg) and a higher stiffness of the gear [83].

Figure 15c reports on the effect of functionally integrated near-contour cooling channels on 3D-printed gears in terms of efficiency and thermal management. In this case, the near-contour cooling helped in reducing the oil supply with regard to immersion lubrication and injection lubrication in the gear. Acceleration losses occur with injection lubrication, caused by the interaction of the injected oil and the squeezing losses because the oil is displaced during tooth contact. The gear was tested experimentally on an efficiency test rig under controlled injection lubrication. A glycose–water mixture was used as the cooling medium. The test results show that near-contour cooling reduced the temperature in the tooth geometry by up to 40 K and improved efficiency. As a result, the oil injection quantity could be significantly reduced, as less oil is required to cool the gear [84]. For the pinion shaft in Figure 15d, a strategy was adopted to cool the gear contour using near-contour liquid-carrying cooling channels in conjunction with ground tooth flanks and ta-C coating, providing optimal effectiveness in dry operation. The gear should then be produced using the 20MnCr5 alloy in the following phase. With a 5% deviation in dry operating efficiency, the goal was to minimise the weight of the gear stage, reduce noise emissions, and extend durability [85].

*3.5. Cost Analysis of Additive Components*

The cost calculation of additive components is very diverse. There are different models for the cost calculation of the components. Some of the diverse costing models are software-based, architecture-based, task-based, and model-based. In these costing models, different criteria are used to calculate the costs, which does not allow for the uniform determination of the actual costs. The cost of restoration, the number of pieces, and the construction effort are often not taken into account [86]. It is, therefore, useful to consider the costs along the product life cycle. In this way, the costs along the process chain of the product life cycle can be discussed individually and then summarised [87]. However, it is often very important to obtain a rough cost estimate of components before the development or the process. For this purpose, widely used manufacturing service providers are a good way to determine the cost of an individual component. By querying several service providers for a component and then averaging the costs, the expected market price can be determined. However, it is not possible to establish a link between the costs and the expected component quality.

**4. Discussion**

In this paper, various electromobility components manufactured using additive manufacturing in the form of 3D metal printing were discussed. The additive components could be grouped into categories such as mechanically stressed or functionally integrated. These categories can also be interconnected. The additive components have advantages over conventionally manufactured components. This is illustrated, on the one hand, by the LeitMot combustion engine [12]. Here, the advantages of additive manufacturing were exploited in many ways. Above all, the lightweight construction, the power density and the

integration of functions through cooling cavities were used. An example of mechanically stressed components is the vehicle frame structure of the Czinger 21C hypercar [80]. Thus, additive manufacturing will be used in the field of electromobility as soon as it offers an advantage over conventionally manufactured components.

A classification scheme for additive manufacturing with regard to an electrified drivetrain can be derived from the findings. The classification includes the basic components with the function of the component, up to the goal of the additive manufacturing measure, such as lightweight construction. The classification scheme is graphically illustrated in Figure 16.

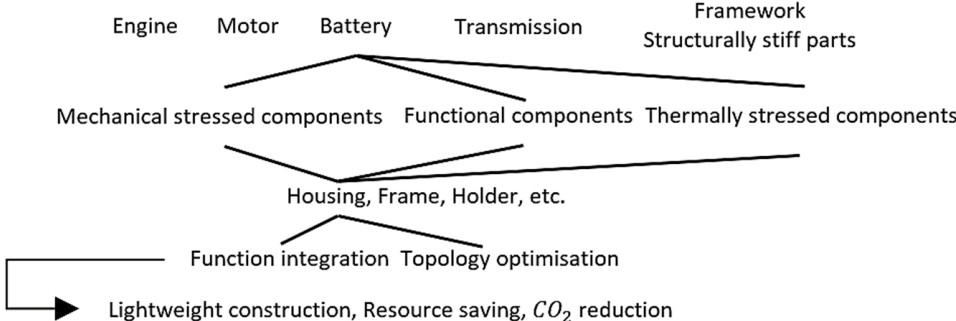

**Figure 16.** Schematic diagram of drivetrain topology with additive manufacturing.

The drivetrain can be subdivided into superordinate categories such as engine → combustion engine, engine → electric motor, or auxiliary motors, batteries, and continuously other drivetrain components. Subcategories representing the superordinate function can then be named. These can be mechanically stressed components or functional components. Then, a component definition is chosen, such as a housing power electronics. Lastly, improvements that are possible through additive manufacturing are added, such as functional integration, and then the objective that is possible through additive manufacturing can be presented, such as empty construction or resource conservation. The exact cost of the components remains difficult [86].

For this reason, the authors recommend averaging the cost of a component by requesting manufacturing from service providers in addition to the costing model used inhouse. Furthermore, the authors recommend systematically carrying out costing along the process chain of the product life cycle.

## 5. Conclusions

This review gives an overview of the current applications of additive manufacturing in the field of electromobility. Additive manufacturing in the field of 3D metal printing has arrived in large parts of the electromobility sector. A wide variety of components are manufactured that are specifically developed for loads and stresses that occur. Components such as electric motors or structural load-bearing components are already available in many applications. There is still potential in the development and use of additive manufacturing for the mobility of tomorrow. Additive manufacturing will continue to progress in the area of function integration, and increase the power density of components. The component quality of additive manufacturing will continue to improve. Due to the improved component quality and mechanical properties of the components, the application areas can be further expanded. There are still deficits with regard to the cost calculation of additive components. The different cost analyses are structured in various ways and do not consider holistic process chains. When producing statements regarding cost calculations, it is necessary to know the structure of the calculation method, which was shown by the present research. It is necessary to promote and publish more scientific work on the subject of additive manufacturing and the mobility of tomorrow.

**Author Contributions:** Conceptualisation, D.S. and C.R., methodology, D.S.; formal analysis, D.S.; investigation, D.S.; resources, D.S.; data curation, C.R.; writing—original draft preparation, D.S. and C.R.; writing—review and editing, D.S. and C.R.; visualisation, M.M. and D.K.H.; supervision, M.M. and D.K.H.; project administration, M.M. and D.K.H..; funding acquisition, M.M. All authors have read and agreed to the published version of the manuscript.

**Funding:** This research was founded by Ministry of Science, Research and the Arts of the State Baden-Wuerttemberg (Germany).

**Institutional Review Board Statement:** Not applicable.

**Informed Consent Statement:** Not applicable.

**Data Availability Statement:** Not applicable.

**Conflicts of Interest:** The authors declare no conflict of interest.

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
