# Peer review of "A Study on Additive Manufacturing for Electromobility"

_wevj, doi:10.3390/wevj13080154_

Round 1

Reviewer 1 Report

- Introduction part is not clear. The new body of the knowledge and scope of the work are not addressed.

- Figure 3 is not clear. Instead of figure, a flow chart depicts various types of AM process has to be included.

- Figure 7 is not clear. The authors need to elaborate the figure.

- it is recommended to provide a separate session to discuss about cost analysis

Reviewer 2 Report

Dear Editor,

The paper entitled 'A study on additive manufacturing for Electromobility' is a review manuscript mainly about  additive manufacturing using electric drive topologies like hybrid and battery-electric vehicles. The topic of the paper has significant importance and it is well organized and written. After following minor revisions, I can propose the paper for publication in your journal. 

-Figure 2 should be replaced with a clearer explanation or different format. 

-The explanation of Figure 3 is not visible. 

-Since the manuscript is a review paper, the introduction part should be improved. The following paper on EV could be helpful.

[1]A comparative study of energy consumption and recovery of autonomous fuel-cell hydrogen–electric vehicles using different powertrains based on regenerative braking and electronic stability control system, Applied Sciences, 2021

[2]Analytical model to predict the extrusion force as a function of the layer height, in extrusion based 3D printing, Additive Manufacturing, 2021

[3]Fuzzy logic and proportional integral derivative based multi-objective optimization of active suspension system of a 4× 4 in-wheel motor driven electrical vehicle, Journal of Vibration and Control, 2022

Author Response

Pleas see the attachment

Round 2

Reviewer 1 Report

Accepted